# Cardiac Glycosides Lower C-Reactive Protein Plasma Levels in Patients with Decompensated Heart Failure: Results from the Single-Center C-Reactive Protein-Digoxin Observational Study (C-DOS)

**DOI:** 10.3390/jcm11071762

**Published:** 2022-03-22

**Authors:** Myron Zaczkiewicz, Katharina Kostenzer, Matthias Graf, Benjamin Mayer, Oliver Zimmermann, Jan Torzewski

**Affiliations:** 1Cardiovascular Center Oberallgäu-Kempten, 87439 Kempten, Germany; katha_789@yahoo.de (K.K.); matthias.graf@klinikverbund-allgaeu.de (M.G.); oliver.zimmermann@klinikverbund-allgaeu.de (O.Z.); jan.torzewski@klinikverbund-allgaeu.de (J.T.); 2Institute of Epidemiology and Medical Biometry, University of Ulm, 89081 Ulm, Germany; benjamin.mayer@uni-ulm.de

**Keywords:** CRP, CRP synthesis inhibition, cardiovascular disease

## Abstract

Recent randomized controlled multi-center trials JUPITER, CANTOS and COLCOT impressively demonstrated the effect of anti-inflammatory therapy on secondary prevention of cardiovascular events. These studies also rapidly re-vitalized the question of whether the C-reactive protein (CRP), the prototype human acute phase protein, is actively involved in atherosclerosis and its sequelae. Direct CRP inhibition may indeed improve the specificity and effectiveness of anti-inflammatory intervention. In the present paper, we report on the final results of our single-center C-reactive protein-Digoxin Observational Study (C-DOS). Methods and Results: Based on the experimental finding that cardiac glycosides potently inhibit hepatic CRP synthesis on the transcriptional level in vitro, 60 patients with decompensated heart failure, NYHA III–IV, severely reduced Left Ventricular Ejection Fraction (LVEF < 40%), and elevated CRP plasma levels were treated by either digoxin + conventional heart failure therapy (30 patients) or by conventional heart failure therapy alone (30 patients). Plasma CRP levels in both groups were assessed for 21 d. Plasma CRP levels on d1, d3 and d21 were compared by regression analysis. CRP levels d21–d1 significantly declined in both groups. Notably, comparative CRP reduction d21–d3 in digoxin versus the control group also revealed borderline significance (*p* = 0.051). Conclusions: This small observational trial provides the first piece of evidence that cardiac glycosides may inhibit CRP synthesis in humans. In case of further pharmacological developments, cardiac glycosides may emerge as lead compounds for chemical modification in order to improve the potency, selectivity and pharmacokinetics of CRP synthesis inhibition in cardiovascular disease.

## 1. Introduction

The elevated plasma levels of C-reactive protein (CRP), the typical human acute-phase protein, predict future cardiovascular events [1]. Whether this is just an epiphenomenon or whether CRP actively contributes to atherogenesis and its sequelae is still controversial [2,3]. Initial experimental studies were suggestive of this [4,5,6]. Genetic analyses from Mendelian randomization trials, however, did not support the concept of CRP being actively involved in human cardiovascular disease [7]. Nonetheless, pitfalls in Mendelian randomization have to be taken into account [8]. Furthermore, species differences in CRP biology limit the value of animal models in this particular area of research [2]. Finally, the experimental data from various laboratories were not accurately evaluated [9] and, unfortunately, have shed negative light on the subject in general. There is, however, international consensus that CRP activates the complement system [10] and binds to Fcγ receptors [11] in atherosclerosis [5,6], and thereby may sustain a chronic inflammatory process in the arterial wall.

Recently, the randomized controlled multi-center trials JUPITER [12], CANTOS [13] and COLCOT [14] impressively demonstrated the effect of anti-inflammatory therapy on secondary prevention of cardiovascular events. These studies also rapidly re-vitalized the question of whether CRP is actively involved in atherosclerosis and its sequelae [3,5], because the effect of anti-inflammatory therapy significantly correlates with CRP reduction in each of these trials. Direct CRP inhibition may indeed improve the specificity and effectiveness of anti-inflammatory intervention.

By using a high throughput screening assay in order to analyze the effect of specific classes of pharmacological agents on CRP transcriptions, in 2010 we showed that endogenous and plant-derived inhibitors of the Na^(+)^/K^(+)^-ATPase, i.e., the cardiac glycosides ouabain, digoxin and digitoxin, inhibit IL-1ß- and IL-6-induced acute phase protein expression in human hepatoma cells and primary human hepatocytes at nanomolar concentrations [15]. Whether this in vitro finding holds true in vivo in humans and is also detected at the CRP plasma level is now being tested in our single-center **C**-reactive protein-**D**igoxin **O**bservational **S**tudy (**C-DOS**) [16,17]. A recently published study showed a significant digitoxin-mediated reduction in CRP levels in mice suffering from sepsis, providing a small piece of evidence that cardiac glycosides are also capable of inhibiting CRP synthesis in mice [18].

The primary aim of this study is to evaluate whether CRP plasma levels can be significantly reduced by digoxin, in addition to optimal medical treatment (OMT) in patients with heart failure and reduced Left Ventricular Ejection Fraction (LVEF) admitted to the hospital with acute cardiac decompensation (NYHA class III and IV).

## 2. Methods

### 2.1. Study Design

Extensive discussions with the Ethical Review Committee of Ulm University, Ulm, Germany, preceded the C-DOS [16,17]. It was initially designed as a blinded, randomized clinical trial comparing two groups (OMT plus digoxin vs. OMT only). The Ethical Review Committee then advised us to change the design, in order to avoid that final medication (digoxin or not) depends on randomization. Cardiac glycoside treatment was recommended to follow clinical needs only, rather than randomization in a study arm. Obeying the Ethical Review Committee’s advice, we designed a prospective observational cohort study, and this design, finally, was approved by the Ethical Review Committee. The patients were recruited in the time span ranging from the end of 2012 until the end of 2019, with an interruption from mid-2016 until mid-2018. This interruption was caused by the surprising pre-emptive review of the study by the Federal Institute for Drugs and Medical Devices of the Federal Republic of Germany (Bundesamt für Arzneimittelsicherheit und Medizinprodukte, BfArM, Bonn, Germany), which examined whether this study was subject to authorization according to the German law ensuring drug safety (Arzneimittelgesetz, AMG; § 4 Abs 23 Satz 1). The review again confirmed that the study was not subject to authorization by the agency, since it was classified by the agency as an observational study on 11 July 2018. This assessment was finally shared by the government of upper Bavaria (Regierung von Oberbayern). Nonetheless, the reglementary process caused an unfortunate interruption in patient recruitment.

### 2.2. Controls/Comparators

The CRP plasma levels of 30 patients with decompensated heart failure, NYHA III and IV, LVEF < 40%, and OMT plus digoxin were compared to the CRP plasma levels of 30 patients with decompensated heart failure, NYHA III and IV, LVEF < 40%, and OMT alone.

### 2.3. Inclusion and Exclusion Criteria, Patient Characteristics

Inclusion criteria: age > 18 years; NYHA III and NYHA IV; acute cardiac failure (acute worsening of dyspnoe, and radiological signs of cardiac congestion); and LVEF < 40% in echocardiography (2 observers, Teichholz/Simpson method). The characteristics of the included patients are summarized in Table 1. Exclusion criteria: significant concomitant disease (acute coronary syndrome, infection, antibiotic therapy, acute renal failure, cancer, and autoimmune disease); CRP > 5 mg/dL, leukocyte count > 12,000/µL, body temperature > 38 °C; and AV-block IIII (for digoxin patients).

Explanation for the exclusion criteria: an exclusion of CRP > 5 mg/dL, leukocyte count > 12,000/µL, body temperature > 38 °C avoids confounding influence of infection. Due to the expected significant proportion of patients that needed to be excluded, approximately 800 patients were assessed for eligibility. Due to the expected [17] final high drop-out rate of 44%, 107 patients were assigned to the trial. A total of 47 patients were disqualified due to various reasons (see Section 3.1), and only 60 patients were finally included in the trial.

### 2.4. Outcome Measures

CRP (and digoxin) plasma levels were assessed at d1, d3, d5, d7 and d21 after inclusion. The primary efficacy endpoint was the comparison of CRP plasma level change between the digoxin and control groups during follow-up (d21–d3). D3 was elected under the assumption that an effect of transcriptional CRP synthesis inhibition by cardiac glycosides on CRP plasma levels may be detectable after 3d of digoxin treatment at the earliest. CRP levels were determined via highly sensitive, particle-enhanced immunological turbidity test assays produced by Roche.

The explanation for the follow-up period: CRP plasma half-life in humans is ~19 h. Therapeutic blood concentration of digoxin, with routine saturation, is reached after d3. We assumed that a potential effect of digoxin on CRP synthesis should be visible on d21.

### 2.5. Methods against Bias

LVEF was echocardiographically analyzed by two investigators via the Teichholz/Simpson method. Multiple regression analysis was used to adjust for potential confounding due to gender, age and cardiac rhythm. CRP plasma levels were assessed by the independent clinical laboratory (Clinic Association Allgäu, Kempten, Germany) via routine CRP measurements. Per protocol analysis of *n* = 60 (30 digoxin vs. 30 control) patients was performed. Blinding was not possible because intervention and control followed clinical needs. Digoxin plasma levels were monitored for safety reasons because of the drug’s small therapeutic window.

### 2.6. Sample Size/Power Calculations

Power calculation was discussed with the Institute of Epidemiology and Medical Biometry of Ulm University. Because no studies exist that investigate the effect of cardiac glycosides on CRP plasma levels, the biometrical classification of the study was “pilot study for subsequent phase III trials”. The sample size of 60 patients in total was evaluated as being adequate to apply the aforementioned multiple regression analysis with 3 confounders. The expected drop-out rate was high due to, for example, the acquirement of lung infection following cardiac decompensation, other infectious diseases or bradycardia due to digoxin treatment. There was no database to conduct a formal sample size calculation due to the lack of retrospective trials in the field.

### 2.7. Feasibility of Treatment

Decompensated heart failure is one of the most common diagnoses on admission in cardiovascular units [19,20]. All admitted patients were screened for inclusion and exclusion criteria. The feasibility and safety of study medication was definitely provided, because cardiac glycosides have been used in cardiac insufficiency for 230 years [21] and, according to the heart failure guidelines, still provide an additive treatment option in NYHA classes III and IV [20,22]. All study participants provided written informed consent.

### 2.8. Statistical Analysis

After the final data acquisition, all variables were descriptively analyzed. The Wilcoxon singed rank test was used to compare the dx-dy differences per group. The Shapiro–Wilk test was used to check the normal assumption. To assess the efficacy of the investigated treatment scheme, multiple regression analysis was performed. The dependent variable was the difference of CRP plasma level at d21 and d3, the independent variables were the group status (digoxin vs. placebo) and the 3 confounding variables gender, age and cardiac rhythm, i.e., sinus rhythm vs. atrial fibrillation. The level of significance was set to 5% (2-sided). The analysis of all secondary endpoints was conducted in an explorative manner. Analyses concerning safety issues were performed by evaluating the adverse events frequencies in both groups. The expected drop-out rate (50%) was high, due to, for example, the potential acquirement of infection during follow-up. Per protocol analysis of *n* = 60 (30 digoxin vs. 30 control) patients was performed.

## 3. Results

### 3.1. Baseline Patient Characteristics, Medication Follow-Up and Drop-Outs

Baseline patient characteristics are summarized in Table 1. The average age was 72.8 years overall with patients in the digoxin group being slightly younger than in the control group (71.8 vs. 73.7 years). A total of 48 men (26 in digoxin group) and 12 women (4 in digoxin group) were enrolled. On admission, 50 patients (26 in digoxin group) were classified as NYHA III and 10 patients (4 in digoxin group) as NYHA IV. LVEF averaged 26.1% in the digoxin group and 24.5% in the control group (25.3% overall). A total of 23 patients suffered from ischemic cardiomyopathy (12 in digoxin vs. 11 in control group), whereas dilated cardiomyopathy (DCM) was diagnosed in 37 patients (18 in digoxin vs. 19 in control group). On admission, ECG showed sinus rhythm in 43 patients (17 in digoxin group) and atrial fibrillation in 15 patients (11 in digoxin group). Atrial flutter and a slow VT were documented in one patient each, both belonging to the digoxin group. Baseline CRP levels on d1 were 1.16 mg/dL in the digoxin group and 0.92 mg/dL in the control group. Further baseline data, including class IA heart failure medication on admission, are shown in Table 1. Class IA heart failure medication is optimized as clinically indicated until d21 in both groups (Table 1) with no statistically significant differences.

Initially, 107 patients were enrolled, of whom 47 patients were not analyzed in this study. This equals a drop-out rate of 43.9%. Patients had to be excluded because of the following reasons: 10 patients (21.7%) withdrew their consent; 9 patients (17.4%) needed antibiotic therapy, which modulates CRP levels itself; 4 (9%) patients needed an ICD implantation, which requests single-shot antibiotic therapy and leads to a rise in CRP levels itself; 3 patients (6.4%) needed other surgical intervention; 2 patients (4.2%) died due to heart failure within the follow-up; and 2 patients (4.3%) were diagnosed with cancer within the follow-up period. Another 2 patients (4.3%) showed complete normalization of LVEF after 21d (main diagnosis: tachycardia-induced cardiomyopathy) and therefore were excluded in line with the study design. The cross over rate was 10.6%, since 5 patients received digoxin due to atrial fibrillation induced tachycardia after they were enrolled in the control group. A total of 5 patients (10.6%) from the digoxin group dropped out due to adverse side effects from digoxin. Another 5 patients (10.6%) were lost in the follow-up; no CRP level could be obtained on d21 after they had been discharged from our hospital.

### 3.2. CRP Levels in the Digoxin Versus Control Group

In the digoxin group, average CRP levels rose from 1.16 mg/dL on d1 to 1.63 mg/dL on d3 and then dropped steadily to 0.54 mg/dL on d21 (see Figure 1A,B). In the control group, a different course of CRP levels was observed, with the corresponding levels being 0.94 mg/dL on d1, 0.98 mg/dL on d3 and 0.72 mg/dL on d21. Detailed CRP levels are shown in Table 2.

As the normal assumption seemed not to be valid, according to the Shapiro–Wilk test, the Wilcoxon signed rank test was performed to compare the within-group differences in the CRP levels. The Wilcoxon ranked sum test was applied in order to compare the difference in the CRP level change (d21–d2) between the digoxin and control group (Table 2). Testing showed a significant decrease in CRP levels within the control as well as within the digoxin group from d1 to d21. The group comparison from d21-d1 between the digoxin and control group revealed no statistically significant differences (Table 2). Notably, a comparison of the CRP plasma level decline d21–d3 between both groups revealed borderline statistical significance (*p* = 0.051; Figure 1; Table 2).

## 4. Discussion

Cardiac glycosides inhibit hepatocellular CRP synthesis in vitro [15]. The aim of C-DOS was to evaluate the effect of cardiac glycosides on human CRP plasma levels in vivo [17]. Digoxin was chosen as an easy-to-handle and easy-to-monitor standard cardiac glycoside for human application [22]. Since therapeutic digoxin plasma levels are commonly reached after 3d of routine saturation and since transcriptional CRP synthesis inhibition likely does not immediately take place following intake, the crucial time points to compare CRP plasma levels were considered to be d21–d3. Digoxin plasma levels in digoxin-treated patients, during this time span, were within the therapeutic range. Due to the standard treatment of heart failure, CRP levels within each group were significantly lower on d21–d1 and d21–d3. Comparing the degree of CRP plasma level reduction for d21–d3 between both groups revealed borderline significance (*p* = 0.051).

### 4.1. Limitiations

C-DOS is a single-center explorative pilot study—no more, no less. It is not a randomized trial. It is not multi-centric. The number of study participants (30 vs. 30 patients) was low. Additionally, due to a 2-year legal re-evaluation (see above), it took a relatively long time (2012–2019) to recruit these patients. The primary endpoint, i.e., CRP plasma level reduction d21–d3, is of borderline significance (*p* = 0.051) only. The drop-out rate of 46% is (as originally expected) high. In addition, a bystander effect of the OMT and recompensation measures on CRP levels cannot be completely excluded with this study design and the relatively low number of patients enrolled.

### 4.2. Possible Implications

C-DOS was designed as a single-center explorative pilot study. Randomization was not permitted by the Ethical Review Committee because medical therapy in this high mortality disease was intended to follow medical necessity rather than admission to a study arm. The number of study participants followed statistical advice (see Section 2.6). Notably, although the number of study participants is low, the primary endpoint, i.e., comparative CRP plasma level reduction from d21–d3 between the groups, revealed borderline significance (*p* = 0.051). This indeed suggests that the significance level may be attained in a larger trial. A high drop-out rate was expected from the start of the trial [16,17]. It was caused mostly by circumstances coinciding with the underlying disease, since a lot of the initially enrolled patients needed either antibiotic therapy (due to relevant infectious disease) or surgery (for example ICD implantation) that could not be postponed during the follow-up period. Compliance after discharge played a role, too, since our hospitals, being inserted into the German healthcare system, have no means of follow-up in the ambulatory setting.

Digoxin was only prescribed when indicated and no contraindications existed [23]. This resulted in a higher number of patients with atrial fibrillation and atrial flutter in the digoxin group. Since it has been shown that atrial flutter and atrial fibrillation both coincide with higher CRP levels [24], this may be a reason for higher baseline CRP levels in the digoxin group on d1. Atrial fibrillation/flutter as a confounding factor also proved to be of borderline significance (*p*-value 0.052) in our statistical analysis.

Statins and ACE inhibitors also lower CRP plasma levels, as has been shown in the JUPITER trial [12] and others [25]. In our observational study, however, there was no statistically significant difference between both groups in terms of ACE-inhibitor and statin therapy. Thus, in our study population, an effect of these two drugs on CRP levels may have been ruled out.

### 4.3. Context of C-DOS

CANTOS [13] showed a 15% relative risk reduction for cardiovascular events by application of the IL-1b specific antibody canakinumab, which subsequently lowered IL-6 and CRP levels. LDL-levels were not influenced. In patients with a high CRP reduction, a subsequent mortality reduction was shown. The LoDoCo trial showed a significantly (5.3% vs. 16%) lower rate of cardiovascular events over the time span of 3 years when patients received low-dose colchicine (0.5 mg/d) in addition to standard therapy [26]. The COLCOT trial [14] showed an analogous effect of colchicine on the secondary cardiovascular event rate in patients post-myocardial infarction. Colchicine inhibits IL-6 synthesis and, consecutively, lowers CRP plasma levels.

Of course, our short-term single-center observational study, in terms of quality, cannot be compared to these controlled randomized multi-center trials, but CRP is the common final stretch in all these trials and may, pathophysiologically, connect them.

Comparing cardiac glycosides to canakinumab, colchicine and recently described CRP apheresis [27,28,29,30], the following issues are crucial: 1. CRP apheresis is highly effective in acute disease, whereas it is likely that CRP synthesis inhibition is not; 2. Canakinumab and CRP apheresis are expensive therapies; 3. Canakinumab in CANTOS has shown immunosuppressive side effects, whereas CRP apheresis in CAMI-1 has not; and 4. Colchicine also is known to cause adverse events.

Thus, drugs with different mechanisms of action at lower costs would certainly reveal new horizons in the primary and secondary prevention of cardiovascular disease. Cardiac glycosides, however, do have a small therapeutic window with significant side effects outside the therapeutic range. Additionally, being Na^(+)^/K^(+)^-ATPase inhibitors, it may be a very long way to develop specific CRP synthesis inhibitors on the structural basis of cardiac glycosides. Consequently, these drugs can only theoretically be used as a pharmacological platform to develop agents that inhibit transcriptional CRP synthesis with less adverse event rates and, ideally, a wider therapeutic range in the distant future. The latter remains challenging.

### 4.4. Conclusions

The aim of C-DOS was to investigate whether digoxin is capable of lowering CRP levels in patients with heart failure and a left ventricular ejection fraction below 40%. A group comparison revealed borderline significance with a *p*-value of 0.051 (d21–d3), likely due to transcriptional CRP synthesis inhibition in vivo. C-DOS, although being a small and observational trial only, provides the first piece of evidence for CRP synthesis inhibition by cardiac glycosides in vivo in humans, and may be a “pilot study for subsequent phase III trials”. A randomized controlled trial investigating the effect of cardiac glycosides on human CRP levels should be the next step, especially to exclude bystander effects. Cardiac glycosides may as well emerge as lead compounds for chemical modification in order to improve the potency, selectivity, and pharmacokinetics of CRP synthesis inhibition in cardiovascular disease.

## Figures and Tables

**Figure 1 jcm-11-01762-f001:**
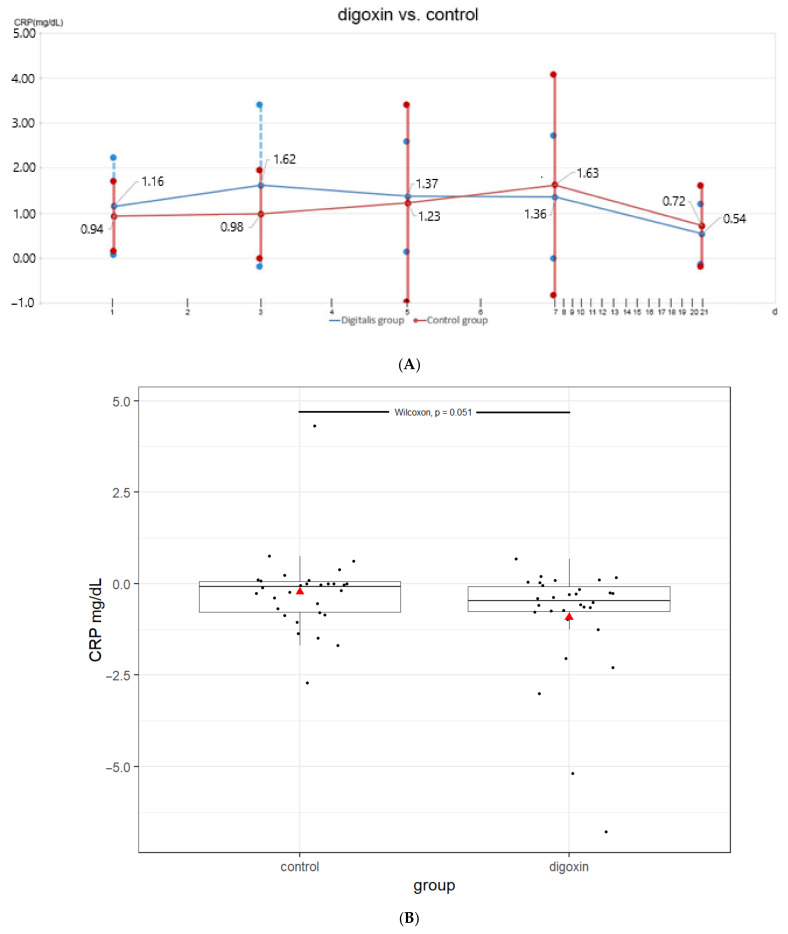
(**A**). Course of mean CRP levels ±SD in the digoxin group and control group (d1 to d21). (**B**) Difference in CRP levels on d21 vs. d3 in the digoxin and control group (box plot: CRP difference d21 vs. d3 digoxin/control group. Triangle = average value, black line = median, upper thin black line = 75. Quantile, lower thin black line = 25. quantile, *p* = 0.051).

**Table 1 jcm-11-01762-t001:** Baseline data and OMT prescribed d1/d21.

Demographic Data	Digoxin Group	Control Group	*p*-Value
Age (±SD)	71.8 years (±10.6)	73.7 years (±8.6)	n.s.
Sex: male/female	26/4	22/8	n.s.
Clinical data			
NYHA III/IV	26/4	24/6	n.s.
LVEF (±SD)	26.1% (±0.08)	24.5% (±0.06)	n.s.
Ischemic cardiomyopathy (total)	12	11	n.s.
Dilated cardiomyopathy (total)	18	19	n.s.
Clinical chemistry			
Digoxin serum level d1 (±SD)	0.28 µg/L (±0.38)		
Digoxin serum level d21 (±SD)	1.46 µg/L (±0.81)		
Creatinine (±SD)	1.20 mg/dL (±0.35)	1.16 mg/dL (±0.35)	n.s.
Sodium (±SD)	140.53 mmol/L (±3.76)	139.00 mmol/L (±3.38)	n.s.
Potassium (±SD)	4.22 mmol/L (±0.43)	4.34 mmol/L (±0.55)	n.s.
Calcium (±SD)	2.27 mmol/L (±0.12)	2.29 mmol/L (±0.88)	n.s.
proBNP (d1)	8484 pg/mL	8528 pg/mL	n.s.
Red blood cell count (d1)	498 million/µL	464 million/µL	n.s.
ECG rhythm at baseline	Number of patients		
Sinus rhythm	17	26	0.010
Atrial fibrillation	11	4	0.037
Slow VT	1	0	n.s.
Atrial flutter	1	0	n.s.
Class IA medication	Number of patients		
b-blocker (d1/d21)	27/27	26/30	n.s./n.s.
ACE inhibitor (d1/d21)	20/22	24/23	n.s./n.s.
AT1 blocker (d1/d21)	5/5	4/5	n.s./n.s.
Aldosterone antagonist (d1/d21)	19/18	17/28	n.s./0.002

Baseline patient characteristics and heart failure medication on d1 and d21 for the digoxin and control groups (comparison by means of independent *t*-test and chi-squared test, respectively). n.s. = not significant.

**Table 2 jcm-11-01762-t002:** CRP levels, normal assumption testing and CRP drop-off comparisons.

	CRP d1 (mg/dL)	CRP d3 (mg/dL)	CRP d5 (mg/dL)	CRP d7 (mg/dL)	CRP d21 (mg/dL)
Digoxin group (±SD)	1.16 (±1.07)	1.62 (±1.80)	1.37 (±1.22)	1.36 (±1.36)	0.54 (±0.67)
Control group (±SD)	0.94 (±0.77)	0.98 (±0.98)	1.23 (±2.19)	1.63 (±2.45)	0.72 (±0.90)
Non-standard distribution	Shapiro–Wilk test	Shapiro–Wilk test			Shapiro–Wilk test
Digoxin group	<0.001	<0.001			<0.001
Control group	<0.001	<0.001			<0.001
	Wilcoxon test	Wilcoxon test	Wilcoxon test		
	Digoxin group	Control group	Group comparison		
d1/d21 (control group)	<0.001	<0.001	0.268		
d3/d21 (control group)	<0.001	0.029	**0.051**		

Average CRP levels and standard deviations on d1, d3, d5, d7 and d21 in both groups, normal assumption testing *p*-values according to Shapiro–Wilk test for both groups; *p*-values (Wilcoxon test) for d1/d21 and d3/d21 CRP level comparison within each group; *p*-values (Wilcoxon test) for group comparison of CRP drop-off from d1/d21 and d3/d21 (***p* = 0.051, borderline significance**).

## Data Availability

Not applicable.

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
