# Peer review of "Cardiac Glycosides Lower C-Reactive Protein Plasma Levels in Patients with Decompensated Heart Failure: Results from the Single-Center C-Reactive Protein-Digoxin Observational Study (C-DOS)"

_jcm, 2022, doi:10.3390/jcm11071762_

Round 1

Reviewer 1 Report

Cardiac glycosides lower C-reactive protein plasma levels in patients with decompensated heart failure - Results from the single-center C-reactive protein-Digoxin Observational Study (C-DOS)

This is an interesting article presenting data from a prospective observational study - 60 patients with decompensated heart failure. The aim of this study was to evaluate, whether C-reactive protein (CRP) plasma levels can be reduced by digoxin in addition to optimal medical treatment in patients with heart failure and reduced left ventricular ejection fraction admitted to the hospital with decompensated heart failure (NYHA class III and IV).The authors explored an important topic in the heart failure population. Due to standard treatment of  heart failure, CRP levels within each group were significantly lower on day 21-day 1 and day21-day3. Comparing the degree of CRP plasma level reduction day21-day3 between both groups revealed borderline significance (p=0.051).

Despite the fact that the statistical significance was not achieved between the analyzed groups and the number of study participants is low, the study will enrich the current knowledge in this field. This analysis may have served as the basis for further research.

The article is well written. Text is clear and ease to read, requiring only a little editing. The conclusions are in line with the evidence and arguments, and address the main question posed.

Congratulations to the authors of their persevere in the implementation of the project.

Author Response

Dear Reviewer No. 1,

thank you for your favorable comments.

Kind regards 

Myron Zaczkiewicz

Reviewer 2 Report

1. What is the main question addressed by the research? Is it relevant and
interesting?   Prior studies have raised the question whether C-reactive protein (CRP), the prototype human acute phase protein, is actively involved in atherosclerosis and its sequelae. Direct CRP inhibition may indeed improve specificity and effectiveness of anti-inflammatory intervention. The single-center study aimed to evaluate effect of digoxin on the changes of CRP levels in heart failure patients.    I am not sure whether it is appropriate to prove digoxin with a direct effect on CRP or a bystander effect.  
2. How original is the topic? What does it add to the subject area compared
with other published material?   The study is original and no prior studies are available.    
3. Is the paper well written? Is the text clear and easy to read? The paper was well written and easy to read.  
4. Are the conclusions consistent with the evidence and arguments
presented? Do they address the main question posed?   The conclusions are limited in heart failure patients.

Author Response

Dear reviewer No 2,

Thank you for your valuable comments, which may have significantly improved our manuscript. Of course, a bystander effect on CRP levels can not be excluded due to the study design, which we clarified as per your suggestion in the limitations section. However, the reason for us to start the C-DOS study was one of our references (Kolkhoff et al. 2011), which indicated that, in vitro, CRP synthesis is inhibited in a Na+/K+ ATP-ase dependent manner by cardiac glycosides.

In addition, we have clarified in the conclusions section that a randomized controlled trial investigating the effect of cardiac glycosides on CRP levels should follow in order to prove glycoside-mediated CRP inhibition in vivo and to exclude bystander effects. We also tried to explain the history of our study more precisely in the introduction section.

Kind regards,

Myron Zaczkiewicz

Reviewer 3 Report

This paper illustrates the results of a small observational trial which evaluated the effects of digoxin in reducing inflammation measured as C-Reactive Protein (CRP). It was carried out on 60 patients’ admitted to hospital with decompensated heart failure NYHA class III-IV and severely reduced left ventricular ejection fraction (<40%). Digoxin was prescribed when indicated in addition to optional medical therapy.

As the authors pointed out, this is a simple, single-centre explorative non-randomised study. The argument investigated is very timely. Recent randomised trails have shown the effects of anti-inflammatory therapy on secondary prevention of cardiovascular events. However, these results are obtained with expensive and/or adverse event-related drugs.

We have some concerns.

  • The Baseline Patient Characteristic regarding haemodynamic (i.e.,: by echography measurement) and biochemical variables (Pro-BNP, High sensitivity CRP, haemoglobin, albumin, N/L ratio,) should be reported to identified the patients better
  • Basal CRP values are within the normal range. Indeed, CRP>5 mg/dl was an exclusion criterion. For this reason, we believe that additional information on the patients should be provided.

Author Response

Dear reviewer No. 3,

thank you for your valuable comments, which may have improved our manuscript significantly.

As per your suggestion, we added BNP levels on day 1 as well as the red blood cell count on day 1 to our baseline data to give more information about the patients at baseline.

In addition, we clarified the assay that was used to measure CRP levels in the material and methods section. The assay used discriminates in the low level range, thus all reported CRP levels are hs-CRP levels.

However, we are not able to provide more baseline data (beyond the above mentioned) for all patients due to the fact that the ethical board of the University of Ulm, Germany, recommended us to design this study as an observational cohort study with the obtaining of routine parameters only, i.e. parameters that would have been obtained in a non-study setting anyway. The parameters presented were the ones obtained for all patients in both groups.

Kind regards

Myron Zaczkiewicz